# Sequential algorithmic modification with test data reuse

**Jean Feng**[1]    **Gene Pennello**[2]    **Nicholas Petrick**[2]    **Berkman Sahiner**[2]    **Romain Pirracchio**[3]    **Alexej Gossmann**[2]

[1]Department of Epidemiology and Biostatistics, University of California, San Francisco
[2]U.S. Food and Drug Administration
[3]Department of Anesthesiology, University of California, San Francisco

## Abstract

After initial release of a machine learning algorithm, the model can be fine-tuned by retraining on subsequently gathered data, adding newly discovered features, or more. Each modification introduces a risk of deteriorating performance and must be validated on a test dataset. It may not always be practical to assemble a new dataset for testing each modification, especially when most modifications are minor or are implemented in rapid succession. Recent work has shown how one can repeatedly test modifications on the same dataset and protect against overfitting by (i) discretizing test results along a grid and (ii) applying a Bonferroni correction to adjust for the total number of modifications considered by an adaptive developer. However, the standard Bonferroni correction is overly conservative when most modifications are beneficial and/or highly correlated. This work investigates more powerful approaches using alpha-recycling and sequentially-rejective graphical procedures (SRGPs). We introduce two novel extensions that account for correlation between adaptively chosen algorithmic modifications: the first leverages the correlation between consecutive modifications using flexible fixed sequence tests, and the second leverages the correlation between the proposed modifications and those generated by a hypothetical prespecified model updating procedure. In empirical analyses, both SRGPs control the error rate of approving deleterious modifications and approve significantly more beneficial modifications than previous approaches.

## 1 INTRODUCTION

Before a machine learning (ML) algorithm is approved for deployment, its performance is usually evaluated on an independent test dataset. If the ML algorithm is modified over time, its performance may change. There are no guarantees on *how* the performance may evolve when the model developer is allowed to introduce modifications in an unconstrained manner. For instance, algorithmic modifications that reduce computational costs may negatively impact model accuracy or precision, and improvements along an aggregate performance metric may come at the cost of lower performance for certain minority subgroups and exacerbate issues of algorithmic fairness. To check that the performance of a proposed modification is acceptable for deployment, the current approach is to run a hypothesis test on a new test dataset, separate from the original one (Feng et al., 2020). The null hypothesis is that the modification is not acceptable; a modification is approved if we successfully reject the null. Nevertheless, large high-quality test datasets are often hard to acquire, particularly in the medical setting.

A major motivation for this work comes from the FDA's recent interest in letting medical device developers update ML-based software, while still ensuring its safety and effectiveness (U.S. FDA, 2019). As outlined in this discussion paper, there are many modifications that may be of interest to software vendors, including the addition/removal of features, changing the model class, and retraining the model on new data. In this work, we consider procedures for testing modifications that are agnostic to the model class and how the modification was trained, e.g. by an online learning algorithm versus a human expert.

When labeled data are expensive and/or difficult to collect, it is tempting to reuse an existing test dataset for determining the acceptability of an algorithmic modification. The danger of test data reuse is that the model developer can learn aspects of the test data when it is used in a sequential and adaptive manner, creating dependencies between algorithmic modifications and the holdout data. For instance,

*Accepted for the 38th Conference on Uncertainty in Artificial Intelligence* (UAI 2022).

the model developer may inadvertently incorporate spurious correlations in the test data to attain over-optimistic performance estimates. This feedback loop introduces bias to the performance evaluation procedure, and adaptively defined hypothesis tests can have drastically inflated Type I error rates (Gelman and Loken, 2017; Thompson et al., 2020).

Recent work tries to protect against inappropriate test data reuse and overfitting by reducing the amount of information released by the testing procedure (Russo and Zou, 2016). The two main approaches are to either coarsen the test outputs along a grid of values (Blum and Hardt, 2015; Rogers et al., 2019) or to perturb the test results with random noise using differential privacy techniques (Dwork et al., 2015a; Feldman and Steinke, 2018). However, existing methods require immensely large datasets to provide protection against overfitting with theoretical guarantees (Rogers et al., 2019). Our aim is to design valid test data reuse procedures for smaller sample sizes that still have sufficiently high power to approve good algorithmic modifications. Our focus is on methods that coarsen the test results. In fact, we consider the extreme case of coarsening where the procedure releases a single bit of information, e.g. whether or not the modification was approved.

When test results are coarsened, the adaptive modification strategy can be described as a tree. As such, one can view test data reuse as a multiple hypothesis testing problem: If we control the family-wise error rate across the entire tree, we control the probability of approving one or more unacceptable modifications. Existing procedures perform a Bonferroni correction with respect to the size of this tree (Blum and Hardt, 2015; Rogers et al., 2019). Nevertheless, the Bonferroni correction is known to be conservative. Instead, we can gain significant power using alpha-recycling (Burman et al., 2009) and accounting for correlation between test statistics (Westfall and Stanley Young, 1993). Indeed, we expect algorithmic modifications to be highly correlated when there is significant overlap between their training data and similarities in their training procedures.

In this paper, we design valid test data reuse procedures based on sequentially rejective graphical procedures (SRGPs) (Bretz et al., 2009, 2011a,b). Although SRGPs are a well-established technique for testing many *pre-specified* hypotheses, many of these procedures cannot be applied when the hypotheses are *adaptively defined in sequence*. The main challenge is that many nodes in the tree of hypotheses are not observed. As such, we introduce two novel SRGPs that are able to account for correlation between adaptively-defined algorithmic modifications without needing to observe these "counterfactual" hypotheses. The first SRGP accounts for correlation between observed nodes in the tree using a fixed-sequence testing procedure. The second SRGP is based on the fact that analysts are not adversarial in practice, i.e. they will not purposefully use prior results to overfit to the test data (Mania et al., 2019; Zrnic and Hardt, 2019). To leverage this fact, this second procedure requires the model developer to pre-specify a hypothetical online learning procedure and then utilizes the similarity between the adaptive and pre-specified modifications to improve testing power. In empirical analyses, both procedures protect against overfitting to the test data, even for small sample sizes, and approve a higher proportion of acceptable modifications than existing approaches. Code is publicly available at https://github.com/jjfeng/adaptive_SRGP.

## 2 PROBLEM SETUP

Suppose the test dataset is composed of $n$ independently and identically distributed (IID) observations $(X_1, Y_1), (X_2, Y_2), \ldots, (X_n, Y_n) \in \mathcal{X} \times \mathcal{Y}$ drawn from the target population. Consider a model developer who adaptively proposes a sequence of $T$ algorithmic modifications $\{\hat{f}_1^{\mathrm{adapt}}, ..., \hat{f}_T^{\mathrm{adapt}}\}$, where each modification is a model that predicts some value in $\mathcal{Y}$ given input $X$. Given criteria for defining the acceptability of a modification, our goal is to approve as many acceptable modifications as possible while controlling the probability of approving an unacceptable modification. Because the decision to approve a modification can be framed as a hypothesis test, a procedure for approving adaptively-defined modifications is equivalent to testing a sequence of adaptively-defined hypotheses $H_1^{\mathrm{adapt}}, ..., H_T^{\mathrm{adapt}}$. Moreover, control of the online family-wise error rate (FWER) in the strong sense, i.e. for any configuration of the null hypotheses, implies control over the rate of approving at least one unacceptable modification.

There are various ways to define acceptability and their corresponding hypothesis test. For example, we may define a modification $\hat{f}$ to be acceptable as long as its expected loss is smaller than that of the original model $\hat{f}_0$. So given a real-valued loss function $\ell$, we would test the null hypothesis

$$H_j^{\mathrm{adapt}} : \mathbb{E}\left(\ell\left(\hat{f}_j^{\mathrm{adapt}}(X), Y\right)\right) \geq \mathbb{E}\left(\ell\left(\hat{f}_0(X), Y\right)\right)$$

at each iteration $j = 1, ..., T$. If we require monotonic improvement in the model performance, we can test if the $j$-th modification is superior to the most recently approved modification $\hat{f}_j^{\mathrm{approved}}(X)$ by testing the null hypothesis

$$H_j^{\mathrm{adapt}} : \mathbb{E}\left(\ell\left(\hat{f}_j^{\mathrm{adapt}}(X), Y\right)\right) \geq \mathbb{E}\left(\ell\left(\hat{f}_j^{\mathrm{approved}}(X), Y\right)\right).$$

Finally, one may also consider multidimensional characterizations of model performance (e.g. model performance within subgroups) and define acceptability as a combination of superiority and non-inferiority tests (Feng et al., 2020). The testing procedures described below only depend on the p-values, so we leave the specific definition of acceptability unspecified until the experimental section.

To limit the adaptivity of the model developer, we consider procedures that sequentially release a single bit of information for each test: One means that the modification is

approved and zero means it is not. Consequently, modifications proposed by any adaptive strategy can be described as a bifurcating tree, where $\hat{f}_{a_t}$ is the *nonadaptive* model tested at time $t$ for the history of approvals $a_t \in \{0, 1\}^{t-1}$, $H_{a_t}$ is the associated *nonadaptive* hypothesis test, and $p_{a_t}$ is its marginal p-value. While one can regard this set of $(2^T - 1)$ hypotheses tests as prespecified, we are only able to observe a specific path along this tree. The unobserved hypotheses are counterfactuals. As such, we need a multiple testing procedure (MTP) that controls the FWER for any adaptively chosen path along the tree *without knowing the exact nature of the counterfactual hypothesis tests*.

A simple approach is to perform a uniform Bonferroni correction for the size of the entire tree. However, the standard Bonferroni procedure has low power because it ignores correlations between models and allocates substantial test mass to hypotheses that are unlikely to be considered. Next we describe procedures that can achieve much higher power.

## 2.1 SEQUENTIALLY REJECTIVE GRAPHICAL PROCEDURES (SRGPS)

We can design more powerful test data reuse procedures by building on *sequentially rejective graphical procedures* (SRGPs), which use directed graphs to define a wide variety of iterative MTPs such as gatekeeping procedures, flexible fixed sequence tests, and fallback procedures (Bretz et al., 2009). SRGPs traditionally assume the set of hypotheses $\{H_j : j \in I\}$ is prespecified and known. The graph initially contains one node for each elementary hypothesis $H_j$, where each node is associated with a non-negative weight $w_j(I)$. The initial node weights, which are constrained to sum to one, control how the total alpha is divided across the elementary hypotheses and correspond to a set of *adjusted* significance thresholds $c_j(I)$. We reject elementary hypothesis $H_j$ in the current graph if its marginal p-value $p_j$ is smaller than $c_j(I)$. For instance, a standard Bonferroni correction is represented by the initial weights of $w_j(I) = 1/|I|$ for all $j \in I$ and significance thresholds $c_j(I) = w_j(I)\alpha$. In addition, the graph contains directed edges where the edge $H_j$ to $H_k$ is associated with weight $g_{j,k}(I)$ for $j, k \in I$ and edge weights starting from the same node must sum to one. When an elementary hypothesis $H_j$ is rejected, its node is removed from the graph and its weight is propagated to its children nodes. This redistribution of test mass, also known as alpha-recycling, increases the power for testing the remaining hypotheses and strictly improves upon simpler procedures that do not use recycling. More specifically, the weight of the edge from $H_j$ to $H_k$, denoted $g_{j,k}(I)$, represents how much of $H_j$'s node weight will be redistributed to $H_k$ if $H_j$ is rejected. So when $H_j$ is removed, the new weight for hypothesis $H_k$ for $k \in I' = I \setminus \{j\}$ is

$$w_k(I') = w_k(I) + g_{j,k}(I)w_j(I). \tag{1}$$

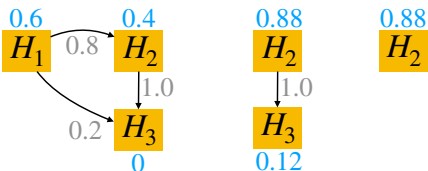

Figure 1: Example sequentially-rejective graphical procedure (SRGP) for hypotheses $\{H_1, H_2, H_3\}$ with initial graph on the left, the middle graph after $H_1$ is rejected, and the right graph after $H_3$ is rejected. Node and edge weights are blue and gray, respectively.

Outgoing edges for all remaining nodes are also renormalized to sum back to one. The SRGP continues until no more hypotheses can be rejected. See Figure 1 for an example.

Assuming all test reports are binary, we can describe adaptive test data reuse as the following *prespecified* SRGP. Let $I_t$ be the set of hypotheses remaining at time $t$, where $I_0$ is the initial set. We only consider SRGPs with nonzero edge weights from hypothesis $H_{a_t}$ to the sequence of hypotheses that would be tested upon rejection of $H_{a_t}$ but prior to the next rejection (though one may also consider more complex recycling procedures). For such SRGPs, the graph of hypotheses has the tree structure seen in Figure 2a, where the only edges in the tree are between hypotheses $H_{a_t}$ and $H_{a_{t'}}$ for $t < t'$ and

$$a_{t',j} = a_{t,j}\mathbb{1}\{j < t\} + \mathbb{1}\{j = t\}. \tag{2}$$

Note that the SRGP tree is *not* the same as the bifurcating tree for generating hypotheses, as the former describes how alpha is recycled.

The model developer must prespecify all initial node weights $w_{a_t}(I_0)$. A simple approach is to perform a uniform Bonferroni correction across all nodes in the graph. We can achieve more power by assigning larger weights to nodes that are more likely to be tested. For example, if the model developer knows that all their modifications will be approved, they should set the initial node weight for $H_{()}$ and all edge weights along the top path in Figure 2a to one.

The model developer will only need to incrementally reveal the edge weights. Let $\tau_t$ denote the time of the latest approval prior to time $t$. At time $t$, the developer must specify the edge weights $g_{a_{\tau_t}, a_t}$ such that the outgoing weights from $a_{\tau_t}$ sum to no more than one. (Note that the edge weight can be treated as a constant because the only relevant edge weight at time $t$ is $g_{a_{\tau_t}, a_t}(I_t)$ and its value is equal to $g_{a_{\tau_t}, a_t}(I_{t'})$ for all $t' < t$.) As such, this procedure for specifying node and edge weights corresponds to a fully prespecified SRGP where a subset of the edge weights are revealed sequentially. To make sure that this SRGP can be executed in the adaptive setting, we must be able to calculate the adjusted significance thresholds for the adaptive hypotheses given the current set of node weights without observing the counterfactual hypotheses.

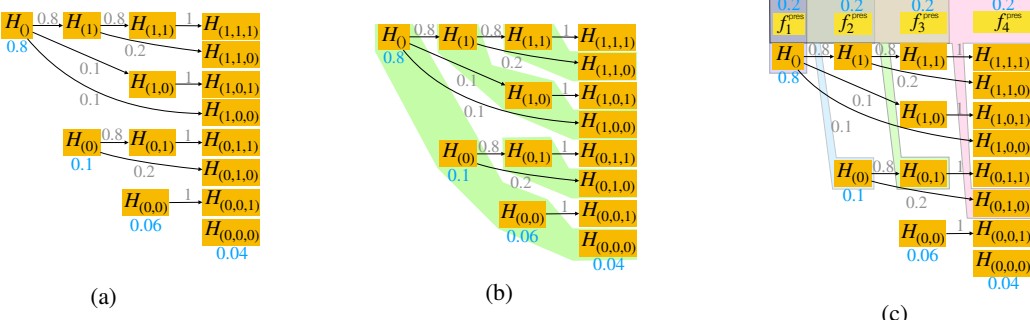

Figure 2: SRGPs for testing $T = 4$ adaptively-defined algorithmic modifications, where $H_{a_t}$ is the hypothesis for testing the adaptive modification given history $a_t \in \{0, 1\}^{t-1}$. (a) A SRGP based on a weighted Bonferroni test. (b) A SRGP that performs a flexible fixed sequence test within each shaded subgroup. (c) A SRGP that adjusts for the correlation between the adaptively proposed modifications and those from a prespecified hypothetical updating procedure, denoted $\{f_t^{\text{pres}} : t = 1, ..., T\}$. As an example, we indicate the correlation adjustments made along the path $H_{()}, H_{(0)}, H_{(0,1)}, H_{(0,1,0)}$.

---

**Algorithm 1** A sequentially rejective graphical procedure (SRGP) that only outputs binary test reports for $T$ adaptive hypotheses given function `compute_sig_threshold`.

**Require:** Initialize $I_0$ as the set of all nodes in the pre-specified tree; initialize $a_1 = ()$ and $\tau_1 = 0$; choose node weights $w_{a_{t'}}(I_0)$ for all $t' = 1, 2, \ldots, T$ and $a_{t'} \in \{0, 1\}^{t'-1}$; and set $w_{a_0}(I_0) = 0$.

**Ensure:** $\sum_{t', a_{t'}} w_{a_{t'}}(I_0) = 1$.

  **for** $t = 1, 2, ..., T$ **do**

    Specify edge weight $g_{a_{\tau_t}, a_t}$ that satisfies outgoing edge weight constraints.

    `# Weight propagation`
    $w_{a_t}(I_t) = w_{a_t}(I_{t-1}) + g_{a_{\tau_t}, a_t} w_{a_{\tau_t}}(I_{\tau_t})$

    **for** all $t', a_{t'}$ such that $H_{a_{t'}} \in I_t$ and $a_{t'} \neq a_t$ **do**

      `# Other weights remain unchanged`
      $w_{a_{t'}}(I_t) = w_{a_{t'}}(I_{t-1})$

    **end for**

    Let $p_{a_t}$ be the marginal p-value from testing $H_{a_t}$.

    Compute significance threshold $c_{a_t}(I_t)$ using `compute_sig_threshold`$(a_t, \{w_{a_{t'}}(I_t) : t', a_{t'}\})$

    **if** $p_{a_t} \leq c_{a_t}(I_t)$ **then**

      Report that $\hat{f}_t^{\text{adapt}}$ has been approved.

      `# Remove node`
      $I_{t+1} = I_t \setminus \{a_t\}$
      $\tau_{t+1} = t$
      $a_{t+1} = (a_t, 1)$

    **else**

      Report that $\hat{f}_t^{\text{adapt}}$ has not been approved.

      $\tau_{t+1} = \tau_t$
      $a_{t+1} = (a_t, 0)$

    **end if**

  **end for**

---

The entire SRGP algorithm for testing adaptive algorithmic modifications is outlined in Algorithm 1. It accepts some function `compute_sig_threshold` that outputs the significance threshold for the adaptively chosen hypothesis given node weights in the current tree. To prove that an SRGP with function `compute_sig_threshold` controls the FWER, we must show that it is a closed test procedure that satisfies the consonance property.

The consonance property is defined as follows. Recall that a closed test procedure uses the following recipe to control the FWER at level $\alpha$: it rejects an elementary hypothesis $H_j$ if the intersection hypothesis $H_K = \cap_{k \in K} H_k$ for every subset $K \subseteq I$ containing the elementary hypothesis $H_j$ is rejected at level $\alpha$ (Lehmann and Romano, 2005). A closed test satisfies the consonance property if the following is true for all $J \subseteq I$: if intersection hypothesis $H_J$ is rejected locally (i.e. its p-value is no more than $\alpha$), there exists some $j \in J$ such that $H_K$ can be rejected locally for all $K \subseteq J$ with $j \in K$ (Gabriel, 1969). In particular, it follows that the corresponding elementary hypothesis $H_j$ can be rejected by the closed test procedure.

When the hypothesis tests are fully prespecified, the consonance procedure lets us perform a closed test using a sequentially rejective (or "shortcut") procedure that iteratively rejects the elementary hypotheses without needing to test every intersection hypothesis (Hommel et al., 2007). So the main benefit of consonance in the non-adaptive setting is that it makes closed testing computationally tractable. In the adaptive setting, the consonance property is even more important because we can perform closed testing *without observing counterfactual or future hypotheses*. That is, the consonance property of an SRGP in the adaptive setting is not simply for computational efficiency, but is necessary for being able to run the MTP at all.

Below, we will describe three SRGPs for testing an adaptive sequence of algorithmic modifications, presented in order of increasing complexity. Each differ in how

`compute_sig_threshold` is defined. To prove that the procedures satisfy the consonance property, it is sufficient to show that the following monotonicity condition holds (Bretz et al., 2009): For every pair of subsets $K, J \subseteq I$ where $K \subseteq J$ and $j \in K$, we have

$$c_j(J) \leq c_j(K). \tag{3}$$

All proofs are provided in the Appendix. The computation time and memory for the proposed SRGPs are all $O(T^2)$. To see this, note that the procedures update no more than $T$ node weights at each iteration.

## 2.2 BONFERRONI-BASED SRGPS

We begin with the simplest SRGP that performs closed testing with a weighted Bonferroni-Holm correction based on node weights, which was originally proposed in Bretz et al. (2009) to test a set of fully pre-specified hypotheses. Nevertheless, this procedure can also be applied in the adaptive setting because the significance thresholds do not depend on observing the counterfactual hypotheses. In particular, this procedure tests the $t$-th adaptive hypothesis given history $a_t$ by comparing its marginal p-value to the corrected significance threshold $c_{a_t}(I_t) = w_{a_t}(I_t)\alpha$. Because this closed test satisfies the monotonicity condition, Algorithm 1 with this significance threshold controls the FWER for the adaptive hypotheses at level $\alpha$.

As a simple example, consider an SRGP that initially assigns Bonferroni-corrected weights to every node and selects nonzero edge weights. This is more powerful than performing a standard Bonferroni correction without any alpha-recycling because the significance thresholds are monotonically non-decreasing at each iteration.

## 2.3 SRGPS WITH FLEXIBLE FIXED SEQUENCE TESTS FOR CORRELATED MODIFICATIONS

When the algorithmic modifications are highly correlated, a Bonferroni-based SRGP will be conservative. We can design more powerful SRGPs by taking into account correlation between the p-values. Bretz et al. (2011a) proposed a procedure that calculates an inflation factor $c(I)$ for intersection hypothesis $I$ such that the probability there exists an elementary hypothesis $H_\mathbb{N}$ with marginal p-value $p_\mathbb{N}$ less than $c(I)w_\mathbb{N}(I)\alpha$, under the null $I$, is no more than $\alpha$. Millen and Dmitrienko (2011) proposed a similar procedure but for test statistics and critical values. Unfortunately, both procedures require knowing the exact correlation structure between all the hypotheses and checking that the monotonicity property holds. This is not feasible in the adaptive setting. To resolve these issues, we propose a new SRGP that (1) partitions the hypothesis tree into sequences of *observed* hypotheses and (2) uses a flexible fixed-sequence test within each subgroup (Huque and Alosh, 2008).

We group together hypotheses that would be tested along a streak of failures immediately following a successful approval (Figure 2b). For each history $a_t \in \{0,1\}^{t-1}$ with $a_{t,t-1} = 1$, we define a subgroup $G_{a_t}$ as the set of hypotheses with histories $a_{t'} = (a_t, \vec{0})$ for any length zero vector, i.e.

$$\left\{ H_{a_{t'}} : a_{t',i} = a_{t,i} \mathbb{1}\{i \leq t-1\}, \forall i = 1, ..., t'-1, \forall t' \geq t \right\}.$$

To test intersection hypothesis $I$, we test each subgroup $G_{a_t} \cap I$ at level $\left( \sum_{H_{a_{t'}} \in G_{a_t} \cap I} w_{a_{t'}}(I) \right)\alpha$. We reject $H_I$ at level $\alpha$ if any of the subgroup-specific tests are rejected. We can show that this controls the Type I error at level $\alpha$ using a union bound. To test a subgroup, we test its hypotheses in the order they are revealed and spend up to the allocated alpha weight. To satisfy the monotonicity property, the significance threshold $c_{a_j}(I)$ for $a_j \in G_{a_t} \cap I$ is defined as the maximum threshold that spends no more than the allocated alpha up to time $j$ for all subsets of hypotheses, i.e.

$$c_{a_j}(I) = \sup \tilde{c}$$
$$\text{s.t. } \Pr\left( p_{a_k} > c_{a_k}(I) \forall a_k \in K, p_{a_j} < \tilde{c} \mid H_{K \cup \{a_j\}} \right)$$
$$\leq \left[ \sum_{\substack{a_k \in ((G_{a_t} \cap I) \setminus K) \\ k \leq j}} w_{a_k}(I) \right] \alpha \quad \forall K \subseteq \{a_k : a_k \in G_{a_t} \cap I, k < j\}. \tag{4}$$

This expression is complicated because it handles arbitrary correlation structures between the p-values. It greatly simplifies in certain cases. For example, if we are performing one-sided Z-tests and the pairwise correlations of the model losses are non-negative, (4) is equivalent to defining $c_{a_j}(I)$ as the solution to

$$\Pr\left( p_{a_k} > c_{a_k}(I) \forall k = t, ..., j-1, p_{a_j} < c_{a_j}(I) \left| \bigcap_{k=t}^{j} H_{a_k} \right. \right)$$
$$= w_{a_j}(I)\alpha.$$

Using the flexible fixed sequence tests from above, we sequentially calculate the significance thresholds and test the adaptive hypotheses. When a hypothesis is rejected, we remove its node and propagate its *local* weight to its children nodes per (1). We can prove the monotonicity condition holds to establish the following result:

**Theorem 1.** *Algorithm 1 with significance thresholds chosen using* (4) *controls the FWER for adaptively defined hypotheses at level $\alpha$.*

## 2.4 SRGPS WITH PRESPECIFIED HYPOTHETICAL MODEL UPDATES

The SRGPs in the above sections protect against the worst case scenario where the model developer is adversarial. In practice, the model developer may have a plan for how

they will update their model over time (i.e. continually refit the model on accumulating data) and will only make small adjustments based on test results. As such, we do not expect the adaptively chosen model at iteration $t$ to stray far from the initial plan. In the most extreme case, we may find that the model developer is not adaptive at all and follows the prespecified procedure perfectly; instead of correcting for $(2^T - 1)$ hypotheses, we would expect that the correction factor to be $O(T)$ instead.

To leverage this similarity assumption, we propose a novel SRGP that requires the model developer to prespecify a procedure for generating hypothetical model updates. This prespecified procedure describes the *exact* steps for how modifications would be generated, e.g. the data stream used, the number of training observations, and hyperparameter selection. These hypothetical model updates are included as additional nodes in the hypothesis graph and assigned positive node weights. Their sole purpose is to improve power for approving the adaptively-defined model updates. These model updates are never formally tested nor approved for deployment. We also do not release *any* information about their test performance, because doing so would increase the amount of information leaked to the model developer and the branching factor of the adaptive tree.

At each iteration, this SRGP constructs a confidence region for the performance of the $t$-th prespecified model update $\hat{f}_t^{\mathrm{pres}}$ by spending its allocated alpha, accounting for its correlation with all prespecified models up to iteration $t-1$. It then tests the $t$-th adaptive model by accounting for its correlation with the prespecified models up to iteration $t$. As such, the power for testing the adaptive modifications increase as their correlation with the prespecified updates increases.

More formally, the critical value and significance threshold at time $t$ are calculated as follows. Let $P_0$ denote the target population and $P_n$ denote the empirical distribution of the test dataset. Here we consider a univariate performance measure $\psi$, where $\psi\left(\hat{f}, P\right)$ is the performance of model $\hat{f}$ with respect to distribution $P$. It is straightforward to extend this procedure to multivariate performance measures (see the Appendix for an example). Denote the deviation between the estimated and true performance as

$$\xi_{t,n}^{\mathrm{pres}} = \psi\left(\hat{f}_t^{\mathrm{pres}}, P_n\right) - \psi\left(\hat{f}_t^{\mathrm{pres}}, P_0\right).$$

For intersection hypothesis $I$, define $\tilde{I}$ as union of $I$ and all prespecified nodes. Define critical value $z_t^{\mathrm{pres}}(I)$ as the largest $\tilde{z}$ such that

$$\Pr\left(\xi_{t',n}^{\mathrm{pres}} > z_{t'}^{\mathrm{pres}}(I)\ \forall t' < t, \quad \xi_{t,n}^{\mathrm{pres}} \le \tilde{z}\right) \le w_t^{\mathrm{pres}}\left(\tilde{I}\right)\alpha. \tag{5}$$

The significance threshold $c_{a_t}(I)$ for testing $H_{a_t}$ is defined

as the largest $\tilde{c}$ such that

$$\Pr\left(\xi_{t',n}^{\mathrm{pres}} > z_{t'}^{\mathrm{pres}}(I)\ \forall t' \le t, \quad p_{a_t} \le \tilde{c}\ \Big|\ H_{a_t}\right) \le w_{a_t}\left(\tilde{I}\right)\alpha. \tag{6}$$

Crucially, these calculations do not depend on observing counterfactual or future hypotheses. Using a union bound, we can show that the Type I error for falsely rejecting the intersection hypothesis $I$ using the critical values defined above is bounded by the sum of the right hand sides of (5) and (6) for all $(t, a_t)$ in $I$. Because the total weight in the graph is always one, we achieve Type I error control at level $\alpha$. Using this idea, we can show that this SRGP indeed controls the FWER:

**Theorem 2.** *Algorithm 1 using significance thresholds defined using equations* (5) *and* (6) *controls FWER at level $\alpha$ for adaptively selected hypotheses.*

## 3 SIMULATION STUDIES

We now present two simulation studies of model developers who adaptively propose modifications to their initial ML algorithm. The developers aim to improve the model's area under the receiver operating characteristic curve (AUC) and quantify the performance increase as accurately as possible. Because our adaptive test data reuse procedures only release a single bit of information at each iteration, we must carefully design the hypothesis tests to obtain a numeric bound on the performance improvement. In particular, we define the $j$-th adaptive hypothesis test as

$$H_{0,j}^{adapt} : \psi\left(\hat{f}_j^{adapt}; P_0\right) \le \psi\left(\hat{f}_0; P_0\right) + \delta_j^{adapt} \tag{7}$$

where $\psi(f, P)$ denotes AUC of model $f$ for distribution $P$ and $\delta_j^{adapt} \ge 0$ is the improvement difference that we are trying to detect. To ensure the model performance tends to improve with each approval, we set $\delta_{j+1}^{adapt} = \delta_j^{adapt} + 0.01$ whenever the $j$-th null hypothesis is rejected. Note that one could consider more complicated hypotheses, each with their pros and cons. For example, one can check that the modifications are strictly improving *and* test for an improvement difference; however, this can be overly stringent.

The purpose of the first simulation study is to investigate FWER control. We do this by simulating a model developer who tries to overfit to the test data based on the information released at each iteration. The purpose of the second simulation study is to investigate power. Here the model developer generally proposes good algorithmic modifications by continually refitting the model given an IID data stream.

In both simulations, we generate $X \in \mathbb{R}^{100}$ using a multivariate Gaussian distribution. $Y$ is generated using a logistic regression model where the coefficients of the first six variables are 0.75 and all other model parameter are zero. The

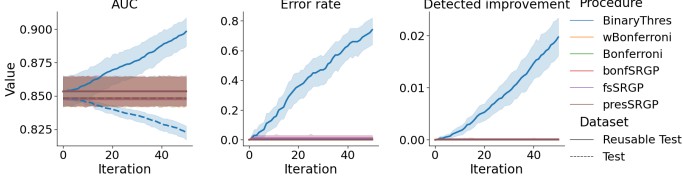

Figure 3: Comparison of multiple testing procedures (MTPs) for approving algorithmic modifications generated by an adversarial model developer. At each iteration, the model developer tests a model that is overfit to the reusable test dataset. We plot the AUCs of the most recently approved model on the reusable test dataset and a completely held out test dataset (left), the rate of incorrectly approving at least one unacceptable modification (middle), and the detected increase in AUC (right). 'Value' is defined in the title of each plot.

modifications are also logistic regression models. We evaluate the two SRGPs proposed in this paper—SRGP with flexible fixed sequence tests (`fsSRGP`) and SRGP with hypothetical prespecified model updates (`presSRGP`)—against relevant baseline comparators, including the standard Bonferroni procedure (`Bonferroni`), a weighted Bonferroni procedure (`wBonferroni`), and the Bonferroni-based SRGP (`bonfSRGP`). For `wBonferroni`, we assigned weights to favor hypotheses with more successful approval histories. In particular, we split the adaptive hypotheses into two groups: those with an approval history with no more than three failed approvals and those with more than three. We then assigned weights such that the hypotheses were equally weighted within each group and the total weight for each group was 0.5. The weights in the SRGPs were defined such that the first outgoing edge (a successful rejection of the hypothesis) is 0.8 and for each subsequent edge, it was assigned 0.8 of the remaining weight. Unless specified otherwise, all the MTPs control the FWER at level $\alpha = 0.1$. Details for deriving test statistics and significance thresholds are provided in the Appendix.

### 3.1 VERIFYING FWER CONTROL

Here we show how MTPs that fail to control the FWER can drastically elevate one's risk of overfitting to the test data, as compared to appropriately-designed adaptive test data reuse procedures. In particular, we consider the naïve procedure that tests every adaptive hypothesis at level $\alpha$ (`BinaryThres`). The reusable test dataset has 100 observations and the model developer tests $T = 50$ modifications. For the purpose of illustration, the initial model is set to the oracle, so all proposed modifications are unacceptable.

The simulated model developer tries to find models that overfit to the test data by searching within the neighborhood of the currently approved model. In particular, the developer

iteratively perturbs the coefficient of each irrelevant variable by 0.6 in the positive and negative directions. When any such modification is approved, the model developer will continue perturbing that coefficient in the same direction until it fails to reject the null hypothesis. For `presSRGP`, the prespecified model update at iteration $t$ is the model with coefficients exactly the same as the initial model except that the coefficient for the $(7 + \lfloor t/2 \rfloor)$-th variable is set to 0.6 if $t$ is even and -0.6 if $t$ is odd.

Figure 3 shows the result from 100 replicates. Notably, `BinaryThres` approves at least one inferior modification with probability 75% and concludes that the modifications by the last iteration improves the AUC by at least 0.02, *even though the AUC actually drops by 0.025 on average*. All the other MTPs appropriately control the FWER at the desired rate of 10% and, thus, protect against over-fitting.

### 3.2 ASSESSING POWER

Here the simulated model developer has access to an IID data stream and iteratively refits a logistic regression model on this data. Because training on more data from the target population tends to improve model performance, the modifications are usually beneficial. However, there is a risk that the modification does not improve performance or that the improvement is negligible, especially because there is a potential for overfitting to the reusable test data set. By testing hypotheses (7), we can restrict approval to only those model updates with meaningful improvements in the AUC.

The test dataset has 800 observations and we allow $T = 15$ adaptive tests. At each time point, the model developer receives a new observation and refits the model. To spend alpha more judiciously, the model developer will only submit the refitted model if the power calculations suggest that the probability for rejecting the null hypothesis exceeds 50%. Specifically, they perform power calculations by setting the true performance improvement to the CI lower bound, which is estimated using split-sample validation. (For simplicity, the power calculations do not perform any multiple testing correction.) To run `presSRGP`, the prespecified model updating procedure also selects updates based on a hypothesis test similar to (7) but replacing the adaptive difference sequence $\delta_j^{\text{adapt}}$ with the prespecified difference sequence $\delta_j^{\text{pres}} = 0.0025(j - 1)$ as well as replacing the adaptive modifications with the prespecified ones.

The procedures differed significantly in power (Figure 4). `presSRGP` performed the best, followed by `fsSRGP`. In particular, the total number of approved modifications and the average AUC of the final approved model were higher when using `presSRGP` compared to `fsSRGP` (p=0.05 and 0.03, respectively). Both `presSRGP` and `fsSRGP` significantly outperformed the other methods ($p \leq 0.005$ for all comparisons).

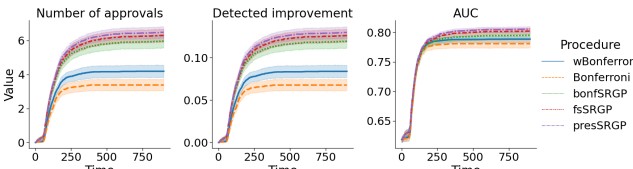

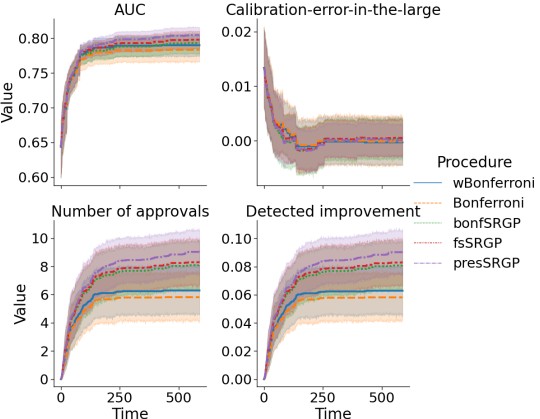

Figure 4: Comparison of multiple testing procedures (MTPs) for approving modifications trained on an IID data stream. A single observation is collected at each time point. The model developer retrains the model on all accumulated data and adaptively decides whether to submit the model for approval. The number of approved modifications (left), the detected increase in AUC (middle), and the AUC of the most recently approved modification (right) are plotted against time.

Figure 5: Comparison of MTPs for approving updates to a gradient boosted tree for predicting acute hypotension episodes. At each time point, model developer observes data from a new admission, retrains the model on all accumulated data, and decides whether to submit the updated model for approval. The test for approval checks that the model's AUC is improving and its calibration-error-in-the-large is close to the ideal value of zero.

# 4 DATA ANALYSIS: PREDICTING ACUTE HYPOTENSION EPISODES

We now apply our procedure for approving modifications to a risk prediction model for acute hypotension episodes (AHEs), one of the most frequent critical events in the intensive care unit (ICU) (Walsh et al., 2013). The ICU is a clinical environment that continuously generates high throughput data. Thus, a model developer can readily collect new data in this setting to retrain an existing model. To mimic this, we use data from the eICU Collaborative Research Database (Pollard et al., 2018). We randomly select 40 admissions to train an initial model, reserve another 500 admissions to construct the reusable test dataset, and use the remaining 589 admissions to simulate a data stream. Data from one admission is observed at each time point, and the model developer is allowed to test a maximum of $T = 15$ modifications.

The task is to predict AHE 30 minutes in advance, where we define AHE as any 5-minute time period where the average mean arterial pressure (MAP) falls below 65 mmHg. The input features to the model are baseline variables age, sex, height, and weight; vital signs MAP, heart rate, and respiration rate at the current time point; and the same set of vital signs five minutes prior. The prediction model is a gradient boosted tree (GBT) and is continually refit on the incoming data.

Here we consider a more complex hypothesis test that checks for calibration-in-the-large (Steyerberg, 2009) *and* improvement in AUC. The $j$-th adaptive null hypothesis is

$$
\begin{aligned}
H_j^{\mathrm{adapt}} : \psi\left(\hat{f}_j^{\mathrm{adapt}}; P_0\right) &\leq \psi\left(\hat{f}_0; P_0\right) + \delta_j^{\mathrm{adapt}} \\
\text{or } E\left[\hat{f}_j^{\mathrm{adapt}}(X) - Y\right] &\notin [-\epsilon, \epsilon],
\end{aligned}
\tag{8}
$$

where $\delta_j^{\mathrm{adapt}}$ is defined using the same procedure as that in Section 3.2, $\hat{f}_j^{\mathrm{adapt}}$ is the modification determined to have sufficient power for rejecting the null, and margin of error $\epsilon$ is 0.05. We will refer to $E\left[\hat{f}_j^{\mathrm{adapt}}(X) - Y\right]$ as calibration-

error-in-the-large. Details on calculating the test statistic and significance thresholds are provided in the Appendix.

Results from 40 replicates are shown in Figure 5. We observe the same ranking of MTPs as that in Section 3.2. The total number of approved models is highest using `presSRGP` compared to the other MTPs ($p \leq 0.005$ for all comparisons). Also, the AUC of the final model approved was also higher using `prespecSRGP` compared to `Bonferroni`, `wBonferroni`, `bonfSRGP`, and `fsSRGP` ($p = 0.002, 0.010, 0.020, 0.015$, respectively). Compared to the previous section, the relative improvement between the methods is smaller because the GBTs improved rapidly at early time points and slowed down thereafter.

To better understand how performance can vary across different datasets, we have included a second data analysis in the Appendix based on a medical imaging task. There, we find that `fsSRGP` and `presSRGP` outperform the rest of the methods, and their approval rates are nearly indistinguishable.

# 5 RELATED WORK

Our paper relates to a large body of work on methods for providing valid statistical inference and preventing false discoveries. Much of this literature has focused on testing prespecified hypotheses on the same dataset while controlling the FWER (Hochberg and Tamhane, 1987; Westfall et al., 2010), false discovery rate (FDR) (Benjamini and Hochberg, 1995), or some variant thereof (van der Laan et al., 2004). More recent work consider testing a sequence

of adaptive hypotheses on prospectively-collected data from a data stream and controlling online error rates (Foster and Stine, 2008; Ramdas et al., 2018). This work considers the setting where we adaptively test hypotheses on the *same* dataset. To control for the bias of reusing the same dataset, testing procedures must limit the amount of information released about the test data (Russo and Zou, 2016). Techniques based on differential privacy, which is a mathematically rigorous formalization of data privacy (Dwork and Roth, 2014), do this by adding random noise (e.g. Laplace or Gaussian noise) to the test statistic or, more generally, the queried result (Dwork et al., 2015a). While theoretical guarantees are available for differential privacy based methods for test data reuse (e.g., (Dwork et al., 2015b; Russo and Zou, 2016; Rogers et al., 2016; Cummings et al., 2016; Dwork et al., 2017; Feldman and Steinke, 2017, 2018; Shenfeld and Ligett, 2019; Gossmann et al., 2021) and others), the required size of the test dataset is prohibitively large for many application domains or require injecting very large amounts of noise (Rogers et al., 2019; Gossmann et al., 2021). An alternative approach is to directly limit the number of bits of information released to the model developer by discretizing the queried result along some grid (Blum and Hardt, 2015). Existing methods essentially perform a Bonferroni correction for the number of distinct hypotheses, which also require unreasonably large test datasets for many applications. To improve testing power, a number of works have assumed that the adaptivity of the model developer is limited (e.g. the models are highly correlated, or the model developer is not entirely "adversarial") to justify the use of a less conservative correction factor (Mania et al., 2019; Zrnic and Hardt, 2019). In contrast, the SRGPs proposed in this work achieve higher power via alpha-recycling and account for the correlation structure without needing to make assumptions about the model developer.

# 6 CONCLUSION

We show how to leverage SRGPs to design valid and powerful approaches for testing a sequence of adaptively-defined algorithmic modifications on the same dataset. The overall steps of this framework are (i) limit the amount of information leakage by reporting only binary test results (approve versus deny modifications), (ii) spend *and recycle* alpha using an SRGP, and (iii) design consonant, closed-testing procedures whose significance thresholds can be computed without needing to observe the counterfactual hypotheses. To account for correlation between the algorithmic modifications, we presented two new SRGPs. `fsSRGP` achieves higher power by leveraging the correlation structure between the observed algorithmic modifications. `presSRGP` asks the model developer to generate a sequence of algorithmic modifications using a prespecified learning procedure and leverages the correlation between the adaptive and prespecified algorithmic modifications. In empirical studies, these

procedures approved more algorithmic modifications than existing methods, with `presSRGP` achieving the highest power.

A limitation of this work is that the model developers were simulated, because there are no publicly available datasets that document modification strategies taken by real-world ML practitioners. In particular, we designed a modification strategy based on one of the most commonly used strategies in the literature: retraining the model on all previously accumulated data (Breck et al., 2017; Amershi et al., 2019). To provide a more complete picture, we have included sensitivity analyses in the Appendix to understand how variations in the model developer's strategy can impact the performance of the proposed methods. Generally, we find that the additional gain in power using `presSRGP` and `fsSRGP` over the other methods depends on the modification strategy. As such, one direction of future work is to design modification strategies that can be paired with these MTPs to maximize power.

Other areas for future research include tuning the node and edge weights in the SRGPs to optimize statistical power and extending the SRGPs to control other error rates (e.g. $k$-FWER and False Discovery Rate) (Robertson et al., 2020). In addition, model developers are often interested in obtaining more detailed test results like p-values and confidence intervals. So an important next step is to design SRGPs that release more information per iteration, perhaps by leveraging differential privacy techniques.

### Acknowledgements

We thank Noah Simon, Charles McCulloch, and Zhenghao Chen for helpful discussions and suggestions. We are grateful to Nicholas Fong for sharing cleaned eICU data.

This work was supported by the Food and Drug Administration (FDA) of the U.S. Department of Health and Human Services (HHS) as part of a financial assistance award Center of Excellence in Regulatory Science and Innovation grant to University of California, San Francisco (UCSF) and Stanford University, U01FD005978 totaling $79,250 with 100% funded by FDA/HHS. The contents are those of the author(s) and do not necessarily represent the official views of, nor an endorsement, by FDA/HHS, or the U.S. Government.

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
