# OpenReview forum: "Sequential algorithmic modification with test data reuse"
_auai.org/UAI/2022/Conference — UAI 2022 Poster_

### Official Review · Reviewer_F5gh · 2022-04-10

**Q2(1) Originality/Novelty:** 3
**Q2(2) Significance/Impact:** 3
**Q2(3) Correctness/Technical Quality:** 3
**Q2(6) Clarity Of Writing:** 3
**Q6 Overall Score:** 7
**Q8 Confidence In Your Score:** 2

**Q1 Summary And Contributions:**

The paper introduces a method to test modifications based on SRGPs, where they define a tree with consonance property so that the adjusted signiﬁcance thresholds can be calculated without observing counterfactual or future hypotheses. They propose two new SRGPs to consider the correlation between the algorithmic modifications. They validate their method on synthetic and real-world data.

**Q2 Assessment Of The Paper:**

More detailed information regarding each of these aspects is given below:

**Q2(4) Quality Of Experiments (Optional):**

3: Good: The experimental evaluation is adequate, and the results convincingly support the main claims.

**Q2(5) Reproducibility:**

3: Good: Key resources (e.g., proofs, code, data) are available and key details (e.g., proofs, experimental setup) are sufficiently well-described for competent researchers to confidently reproduce the main results.

**Q3 Main Strengths:**

- The authors have included a good literature survey on similar problems in previous published work.
- The authors reduce the adaptive modiﬁcation strategy as a tree and leverage consonance property to compute the thresholds without needing to observe the counterfactual hypotheses, which seems to be novel.
- The authors attempt to incorporate prior knowledge by pre-specifying a procedure for generating hypothetical model updates.


**Q4 Main Weakness:**

- Some detailed descriptions may need to be refined, see Q5.
- Some properties of the proposed algorithm such as computational complexity need to be added.


**Q5 Detailed Comments To The Authors:**

- For the proposed presSRGP algorithm, will the performance heavily depend on the quality of the prespecified model updates?
- The authors mention that the Bonferroni correction is overly conservative. Will the performance of Bonferroni correction be better if tune the value of alpha properly?
- What is the computational complexity of the proposed algorithm?


**Q7 Justification For Your Score:**

The authors shed light on a new perspective to analyze test data reuse by reducing the adaptive modiﬁcation strategy as a tree and leveraging consonance property to compute the thresholds without needing to observe the counterfactual hypotheses, which seems to be novel. They also attempt to incorporate prior knowledge into this process, which is believed to be interesting to the community.

**Q9 Complying With Reviewing Instructions:**

1: Yes.

---

### Official Review · Reviewer_XLqJ · 2022-04-12

**Q2(1) Originality/Novelty:** 3
**Q2(2) Significance/Impact:** 2
**Q2(3) Correctness/Technical Quality:** 3
**Q2(6) Clarity Of Writing:** 2
**Q6 Overall Score:** 6
**Q8 Confidence In Your Score:** 2

**Q1 Summary And Contributions:**

Machine learning models are deployed in production and often require constant adjustments. However, adjustment can lead to performance degradation, so reasonable detection methods are needed to determine whether the adjustment  can be deployed.
This paper proposes valid test data reuse procedures based on sequentially rejective graphical procedures (SRGPs) The methods proposed is model-agnostic and suitable for smaller sample sizes.
Experiments are conducted to support the proposed methods.

**Q2 Assessment Of The Paper:**

More detailed information regarding each of these aspects is given below:

**Q2(4) Quality Of Experiments (Optional):**

3: Good: The experimental evaluation is adequate, and the results convincingly support the main claims.

**Q2(5) Reproducibility:**

3: Good: Key resources (e.g., proofs, code, data) are available and key details (e.g., proofs, experimental setup) are sufficiently well-described for competent researchers to confidently reproduce the main results.

**Q3 Main Strengths:**

The adjustment of the deployed models is a significant problem.
- This paper provides more detailed experimental details.
- The experimental results support the method proposed in this paper to a certain extent.


**Q4 Main Weakness:**

The writing in section 2 is a little confusing.
- Experiments need more real data sets to support conclusions.
- Most modifications may not be beneficial and/or highly correlated in practice.


**Q5 Detailed Comments To The Authors:**

Maybe you can combine the two proposed methods to make a greater contribution.
- Maybe you can conduct experiments on more real datasets.
- Maybe you can provide a notation table in section 2.

**Q7 Justification For Your Score:**


The problem studied by this paper is important for ML in production. These methods may be more practical for production. And experiments support the claims to some extent. Although the writing is a little confusing and the techniques may be promoted,  this paper deserve "weak accept".


**Q9 Complying With Reviewing Instructions:**

1: Yes.

---

### Official Review · Reviewer_sLXf · 2022-04-13

**Q2(1) Originality/Novelty:** 2
**Q2(2) Significance/Impact:** 2
**Q2(3) Correctness/Technical Quality:** 3
**Q2(6) Clarity Of Writing:** 2
**Q6 Overall Score:** 5
**Q8 Confidence In Your Score:** 3

**Q1 Summary And Contributions:**

In this paper, the authors suggest a way of testing the impact of modifications to machine learning algorithms. In particular, the proposed approach is for checking whether modifications impact the overall performance of a trained model in case of algorithmic changes. The approach is based on statistical hypothesis testing, which seems to be an appropriate choice.

**Q2 Assessment Of The Paper:**

More detailed information regarding each of these aspects is given below:

**Q2(4) Quality Of Experiments (Optional):**

3: Good: The experimental evaluation is adequate, and the results convincingly support the main claims.

**Q2(5) Reproducibility:**

3: Good: Key resources (e.g., proofs, code, data) are available and key details (e.g., proofs, experimental setup) are sufficiently well-described for competent researchers to confidently reproduce the main results.

**Q3 Main Strengths:**

+ A very much interesting practical problem
+ An interesting theoretical solution that seems to improve previous work
+ A comparison of various techniques using simulated cases


**Q4 Main Weakness:**

- The evaluation is only based on simulation and not on real-world data
- The details behind the problem are not sufficiently discussed. It remains unclear whether the model or the algorithm is modified.


**Q5 Detailed Comments To The Authors:**

The authors nicely introduce the problem. The problem itself seems also to be recent and very interesting for practical applications. In the introduction (and also later on) it is a little bit unclear what modifications are considered (on side of the model via retraining or the algorithm). It seems that the paper considers algorithm (or implementation) changes and their impact on the performance of the (learned) model. Hence, the suggestion is to improve the discussion to clearly state which changes are considered.

In addition, the paper only includes a study that is based on simulation results. This might be appropriate to an extent. The evaluation at least also considers other approaches. However, it would have been better to evaluate the approaches on real-world data to show their practical usefulness. At the moment the work seems to be rather theoretical and not that applicable. At least the authors should discuss limitations in detail.


**Q7 Justification For Your Score:**

The considered problem seems to be an interesting one for practical applications. The proposed approach is reasonable and there is also a simulation-based evaluation included. Unfortunately, there is no evaluation based on real-world data. The applicability in a practical setting might be given but maybe also a simpler method would be sufficient. In addition, there are some aspects that remain unclear after reading the paper. A more precise discussion should be provided.

**Q9 Complying With Reviewing Instructions:**

1: Yes.

---

### Official Review · Reviewer_zdBV · 2022-04-14

**Q2(1) Originality/Novelty:** 3
**Q2(2) Significance/Impact:** 2
**Q2(3) Correctness/Technical Quality:** 3
**Q2(6) Clarity Of Writing:** 3
**Q6 Overall Score:** 6
**Q8 Confidence In Your Score:** 4

**Q1 Summary And Contributions:**

The authors consider the problem of gradual improvement of models without using a fresh test set for each considered improvement (to avoid overfitting). They design novel procedures for test data reuse based on SRGPs: with fixed sequence tests and with hypothetical prespecified model updates. They evaluate their procedures using two simulation experiments, and on real-world data, and show their methods outperform the baseline procedures based on Bonferroni correction.

**Q10 Ethical Concerns (Optional):**

There are no ethical concerns.

**Q2 Assessment Of The Paper:**

More detailed information regarding each of these aspects is given below:

**Q2(4) Quality Of Experiments (Optional):**

2: Fair: The experimental evaluation is weak: important baselines are missing, or the results do not adequately support the main claims.

**Q2(5) Reproducibility:**

3: Good: Key resources (e.g., proofs, code, data) are available and key details (e.g., proofs, experimental setup) are sufficiently well-described for competent researchers to confidently reproduce the main results.

**Q3 Main Strengths:**

- The paper is mostly clearly written.
- The proposed procedures are justified and have a clear motivation.
- The mathematical foundation seems sound.
- The proposed procedures outperform the baselines.


**Q4 Main Weakness:**

- The abstract does not adequately tell the reader what the paper is about and needs to be revised. The end of the Introduction section is not helpful and would benefit a lot of signposting.
- All figures have unclear axis labels: all y-axes are labelled "Value", and several plots have "Time" as their x-axis without a clear unit (second? epochs?).
- Section 3.2. seems incomplete. You are stating the numbers means but it's not clear that the difference is significant between fsSRGP, bonfSRGP and fsSRGP. The same is true for Section 4. I would like to see a discussion whether the improvement is statistically significant.

**Q5 Detailed Comments To The Authors:**

- "Recents works" --> "Recent work" (several places)
- Please clarify what it means for a modification to be acceptable for deployment in the Introduction.
- Insert commas between $(X_1,X_2)$ and $(X_2,Y_2)$.
- The defition of hypotheses $H_j^{\mathrm{adapt}}$ and $H_0^{\mathrm{adapt}}$ have the inequality sign flipped: they should have $\leq$ instead of $\geq$.
- It is easy to miss the definition of the consonance property in the left column of Page 4, please emphasise it better and break up this paragraph in some way.
- Use $\bigcap$ for intersections of sequences.
- Spacing issues in the formula in the right column on Page 5 and in the Appendix: there is no spacing before a quantifier.
- "To spend alpha more judiciously, the model developer will only submit the refitted model if the power calculations suggest that the probability for rejecting the null hypothesis exceeds 50%." This does not seem justified. Could you please clarify why you don't want a higher probability of rejecting the null hypothesis?
- Use \mathit{} for italic superscripts on Pages 7 and 8.
- Consistency: "adapt" in superscript should be either italic or roman, but it should be consistent.
- Algorithm 1: the $I_{t+1}$ definition near the end of the algorithm should have curly braces around $a_t$.

**Q7 Justification For Your Score:**

The paper is mostly fine, but the comparison between methods seems incomplete and the abstract and Introduction could be more helpful to the reader.

**Q9 Complying With Reviewing Instructions:**

1: Yes.

---

### Decision · Program_Chairs · 2022-05-15

**Decision:**

Accept (Poster)

**Comment:**

Meta Review: This paper proposes an interesting solution for reusing the testing dataset while preventing the overfitting issue. All the reviewers reach a consensus on the acceptance. The authors are encouraged to take all the comments into consideration and further improve the paper in the camera ready.